# Ethanol treatment for sterilization, concentration, and stabilization of a biodegradable plastic–degrading enzyme from *Pseudozyma antarctica* culture supernatant

**Takumi Tanaka**[⊙], **Ken Suzuki**[⊙], **Hirokazu Ueda, Yuka Sameshima-Yamashita, Hiroko Kitamoto** *

Institute for Agro-Environmental Sciences (NIAES), National Agriculture and Food Research Organization, Kannondai, Tsukuba, Ibaraki, Japan

⊙ These authors contributed equally to this work.
* kitamoto@affrc.go.jp

**Data Availability Statement:** All relevant data are within the paper and its Supporting Information files.

## Abstract

Biodegradable plastics must be sufficiently stable to maintain functionality during use but need to be able to degrade rapidly after use. We previously reported that treatment with an enzyme named PaE, secreted by the basidiomycete yeast *Pseudozyma antarctica* can speed up this degradation. To facilitate the production of large quantities of PaE, here, we aimed to elucidate the optimal conditions of ethanol treatment for sterilization of the culture supernatant and for concentration and stabilization of PaE. The results showed that *Pseudozyma antarctica* completely lost its proliferating ability when incubated in ≥20% (v/v) ethanol. When the ethanol concentration was raised to 90% (v/v), PaE formed a precipitate; however, its activity was restored completely when the precipitate was dissolved in water. To reduce ethanol use, PaE was successfully concentrated and recovered by sequential ammonium sulfate precipitation and ethanol precipitation steps. Over 90% of the activity in the original culture supernatant was recovered and the specific activity was increased 3.4-fold. By preparing the enzyme solution at a final concentration of 20% (v/v) ethanol, about 60% of the initial activity was maintained at ambient temperature for over 6 months without growth of microbes. We conclude that ethanol treatment is effective for sterilization, concentration, and stabilization of PaE, and that concentrating PaE by sequential ammonium sulfate precipitation and ethanol precipitation substantially increases the PaE purity and decreases ethanol use.

## Introduction

Since large-scale production of the plastics started in the 1950's, more than 8 billion tons of plastic products have been used worldwide, and more than 6 billion tons have been disposed of as waste [1, 2]. In recent years, the replacement of conventional (non-biodegradable)

**Funding:** This work was supported by a grant from the "Science and Technology Research Promotion Program for Agriculture, Forestry, Fisheries, and Food Industry" of the Ministry of Agriculture, Forestry and Fisheries (Japan) (https://www.maff.go.jp/) (grant number, 25017AB; for HK). This work was also supported by the "Research Program on Development of Innovative Technology" of the Bio-oriented Technology Research Advancement Institution, National Agriculture and Food Research Organization (Japan) (http://www.naro.affrc.go.jp/) (grant number, 01029C; for HK). The funders had no role in study design, data collection and analysis, decision to publish, or preparation of the manuscript.

**Competing interests:** The authors have declared that no competing interests exist.

plastics with biodegradable plastics (BPs), which are degraded in natural environment, has been strongly promoted worldwide [3–5]. However, when BP products are designed to be sufficiently stable to maintain their functionality throughout the period of use, their degradation rate in an uncontrolled environment is often slower than desirable [6–8]. Without a means of rapidly degrading BPs, there is concern that they may remain and accumulate in the environment where they are used.

Treatment with a BP-degrading enzyme can accelerate the degradation of BPs; various types of such enzymes have been isolated from microbia [2, 9–11]. We previously discovered that a basidiomycetous yeast, *Pseudozyma antarctica* (currently *Moesziomyces antarcticus*), isolated from the surfaces of rice husks strongly degrades BPs [12]. This yeast secretes a cutinase-like enzyme, named PaE, which can efficiently degrade BPs such as polybutylene succinate, polybutylene succinate-*co*-adipate, polycaprolactone, and amorphous polylactic acid [12, 13]. When 1-cm$^2$ of commercial BP agricultural mulch film was treated with diluted *P. antarctica* culture supernatants, which contained 1-unit of PaE, the film was rapidly degraded and broke down within 72 h [14]. This high BP-degrading activity means that PaE is potentially a practical enzyme for the degradation of used BP products. To reduce the production cost of PaE, recombinant *P. antarctica* strains secreting large amounts of PaE have been constructed [15, 16].

If typical existing microbial culture facilities can be used for enzyme production without additional capital investment, PaE can be produced at a low cost. In light of the Cartagena Protocol on Biosafety (http://bch.cbd.int/protocol/), to eliminate potential environmental effects of the enzyme solution, the cultured recombinant microbes should not be allowed to leak into the external environment. However, filtration through a cell-impervious membrane to completely remove the cultured microbes is expensive and may be difficult to apply to large volumes of culture supernatant in some existing facilities. Thus, there is a need for alternative methods to prevent both proliferation of cultured microbes and contamination by foreign spoilage microbes. In addition, the enzyme solution must be concentrated for transport, and the enzyme activity must be maintained until use. Here, we attempted to find the optimal conditions of ethanol treatment for sterilization of the culture supernatant and for concentration and stabilization of PaE from *P. antarctica*.

## Materials and methods

### Strains, media, and culture conditions

The wild-type strain *Pseudozyma antarctica* GB-4(0) was originally isolated from rice husks [12] and deposited in the National Agriculture and Food Research Organization Genebank, Japan (accession no. MAFF 306999). A transformant strain with high PaE productivity, *P. antarctica* L1-S12, was constructed from the wild-type strain by introducing an *Eco*RI-digested PaE-producing plasmid pPaLYS12XG-PaCLE1 [16] and stored in our laboratory. For maintenance and culture of the strains, minimal medium (0.17% yeast nitrogen base without amino acids and ammonium sulfate [BD Bioscience, New Jersey, USA], 0.5% ammonium sulfate, and 2% glucose), YM medium (0.3% yeast extract, 0.3% malt extract, 0.5% peptone, and 1% glucose), and YPD medium (0.5% yeast extract, 2% peptone, and 1% glucose) were used with 2% agar when necessary, after autoclaving at 121°C for 20 min. Petri dishes (9-cm diameter) were used for the agar plates. PaE was produced in a jar fermentor as previously described [15].

### Inactivation of *P. antarctica* cells by ethanol

The two strains of *P. antarctica*, GB-4(0) and L1-S12, were separately cultivated with 3 mL of YPD medium in a test tube at 30°C for 24 h by reciprocal shaking at 150 rpm. Cells were

counted on a hemocytometer, and the cultures were diluted to $1 \times 10^7$ cells/mL with 0.9% (w/v) NaCl. Various amounts of ethanol (0–900 μL) were added to 100 μL of the diluted cultures, and each suspension was immediately adjusted to 1 mL with 0.9% NaCl, to give final ethanol concentrations ranging from 0% to 90% (v/v). After vortexing, the suspensions were incubated at 4°C or 30°C for 1, 2, 4, 24, or 48 h, or 7 days. Cells were then collected by centrifugation at 17,400$g$ for 1 min, washed with 0.9% (w/v) NaCl, and resuspended in 100 μL of 0.9% NaCl. An aliquot of each resuspension (5 μL) containing $5 \times 10^4$ cells was spotted on YM agar plates and then cultivated at 30°C for 3 days. The same-sized aliquot was also inoculated into 800 μL of YM liquid medium in a 96 deep-well plate and cultivated at 30°C with shaking at 1,500 rpm in a Maximizer MBR-022UP incubator shaker (TAITEC, Saitama, Japan) for 3 days. After cultivation, the growth of *P. antarctica* was confirmed based on its ability to form colonies on the YM agar plates and an increase in turbidity at the optical density in light wavelength 660 nm (OD$_{660}$) of YM liquid medium.

## Degrading activity of PaE against polybutylene succinate-*co*-adipate

The BP-degrading activity of PaE was measured as described in Shinozaki et al. (2013) with the following modifications: 0.09% (w/v) emulsified polybutylene succinate-*co*-adipate (PBSA) (Bionolle EM-301, Showa Denko K. K., Tokyo, Japan) was used as the substrate in 25 mM Tris–HCl buffer (pH 9.0), and 10 μL of the enzyme solution was used in a total volume of 2 mL. After incubation at 30°C, the reduction of OD$_{660}$ with time was measured. One unit of emulsified PBSA-degrading activity was defined as a reduction of 1.0 OD$_{660}$ unit per min in the reaction mixture at a 10-mm light path length.

## Protein quantification

Crude protein concentrations were determined by using Quick Start Bradford 1× Dye Reagent (Bio-Rad Laboratories, Inc., Hercules, CA, USA). Each sample solution (20 μL) was mixed with 1 mL of dye reagent and incubated for 15 min, and then absorbance at 595 nm was measured. A standard curve was prepared by using a Quick Start Bovine Gamma Globulin Standard Kit (Bio-Rad Laboratories, Inc.). The concentration of purified PaE was calculated as described previously [13]. In short, the purified PaE solution was appropriately diluted, and its absorbance was measured at 280 nm (A$_{280}$). Then, the concentration of PaE was calculated according to the following equation: A$_{280}$ ($M^{-1}$ cm$^{-1}$) = 5,500n$_W$ + 1,490n$_Y$ + 125n$_C$ [17, 18], where $M$ (= 20,362.41) is the average molecular weight calculated using GENETYX software ver. 9 (GENETYX, Tokyo, Japan), n$_W$ is the number of Trp residues (= 1), n$_Y$ is the number of Tyr residues (= 7), and n$_C$ is the number of disulfide bonds (= 2), per polypeptide chain based on the mature amino acid sequence of PaE [13].

## Ethanol precipitation

Culture filtrates of *P. antarctica* L1-S12 were obtained from jar-cultured broth by centrifugation and filtration (pore size, 0.45 μm). Ethanol was added to each culture filtrate to a final concentration of 0%, 25%, 50%, 70%, or 90% (v/v), and the filtrates were incubated overnight at ambient temperature. The generated precipitate from each filtrate was collected by centrifugation at 10,000$g$ for 10 min and then dissolved in deionized distilled water (DDW) to the same volume as that of the initial culture filtrate. Ethanol was added to a final concentration of 70%; the precipitate formed was removed by centrifugation at 10,000$g$ for 10 min. Ethanol was added to the supernatant to a final concentration of 90% (v/v). The generated precipitate was again collected by centrifugation at 10,000$g$ for 10 min and dissolved in DDW to the initial volume.

## Combination of ammonium sulfate precipitation and ethanol precipitation

The procedure for concentrating PaE from the culture filtrate is shown in Fig 1A. All operations were performed at ambient temperature. Ammonium sulfate (314 mg) was dissolved in 1 mL of *P. antarctica* L1-S12 culture filtrate to afford 50% saturation. The generated precipitate was collected by centrifugation at 10,000*g* for 10 min. The precipitate was suspended in 1 mL of 70% (v/v) ethanol and then the supernatant was recovered by centrifugation at 10,000*g* for 10 min. Ethanol (2 mL) was added to the supernatant to give a final ethanol concentration of 90% (v/v); then, the generated precipitate was collected by centrifugation at 10,000*g* for 10 min and dissolved in 1 mL of DDW.

The protein pattern of each fraction was checked by sodium dodecyl sulfate–polyacrylamide gel electrophoresis (SDS-PAGE) using a Tris-tricine buffer system (e-PAGEL E-T15S and EzRun T, ATTO, Tokyo, Japan) with Precision Plus Protein Unstained Standards (Bio-Rad) as the molecular weight markers. Protein bands were visualized by silver staining (Silver Staining Kit Protein, GE Healthcare Life Sciences, Buckinghamshire, England).

## Stability of PaE in various ethanol concentrations

To examine the stability of PaE in ethanol solution, we added ethanol to culture filtrates of *P. antarctica* L1-S12 to final concentrations of 0, 10, 25, 50, 75, and 90% (v/v), and stored them at ambient temperature for up to 206 days. During this period, the residual PBSA-degrading activities over time were monitored.

The stability of the concentrated PaE dissolved in a final concentration of 20% (v/v) ethanol was examined for 230 days under two temperature conditions (4°C and ambient temperature). The residual PBSA-degrading activities were monitored.

# Results

## Inactivation of *P. antarctica* cells by ethanol

The cell viabilities of *P. antarctica* strains GB-4(0) and L1-S12 under various combinations of ethanol and temperature conditions were measured as (a) the colony forming ability on YM agar plate and (b) the increment of cell turbidity in YM liquid culture (Table 1). No viable yeast cells were detected in cell suspensions of GB-4(0) or L1-S12 following incubation in the presence of ≥25% (v/v) ethanol at 4°C for 1 h or ≥20% (v/v) ethanol at 30°C for 1 h. The lower the ethanol concentration and the lower the incubation temperature, the longer the survival time. In the presence of 10% ethanol, both GB-4(0) and L1-S12 cells died by 48 h at 30°C, but survived for 7 days (168 h) when stored at 4°C. When the ethanol concentration was increased to 20% (v/v) at 4°C, the cells of GB-4(0) and L1-S12 died by 24 and 48 h, respectively. Detection of viable cells was more sensitive in liquid culture than in plate culture.

## Stability of PaE in culture filtrates with ethanol

The temporal change of PaE activity, measured as PBSA-degrading activity, in culture filtrates of *P. antarctica* L1-S12 containing various concentrations of ethanol is shown in Fig 2A. In the culture filtrates containing 10% or 25% (v/v) ethanol at ambient temperature over 200 days, the activity remained at 67% or 60%, respectively, of the original level (0 days, 0% ethanol). Without ethanol, the activity rapidly declined within 60 days and was completely lost by 200 days, presumably due to severe spoilage-microbe contamination. When the ethanol concentration was ≥50%, the enzyme activity in the culture filtrate decreased to less than 50% after 200 days. The residual activities decreased at higher ethanol concentrations even though no microbial contamination was observed (Fig 2A).

(a)

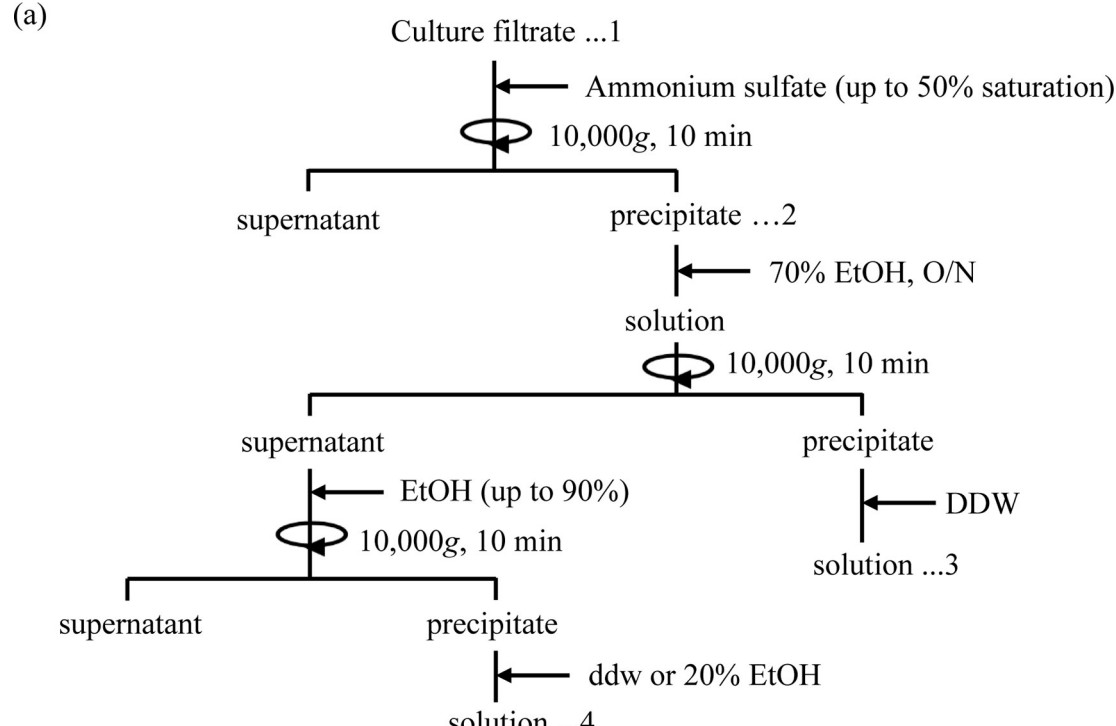

(b)

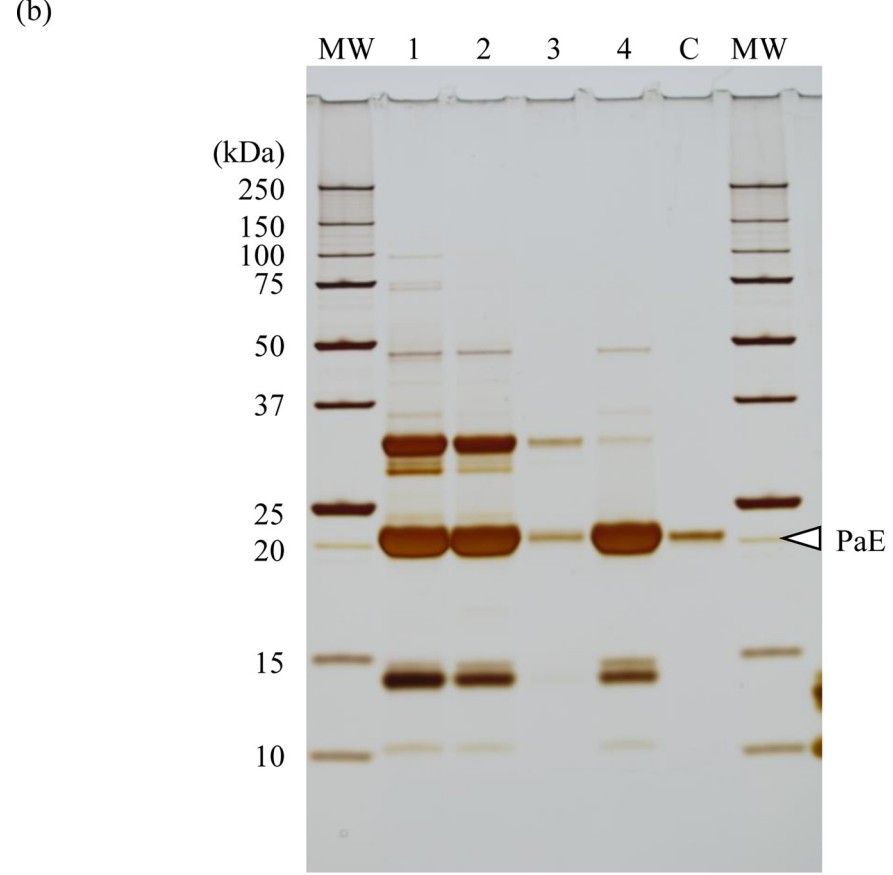

**Fig 1. Concentration of PaE by precipitation with ammonium sulfate and ethanol.** (a) Schematic of the concentration steps. The numbers correspond to the lanes in the SDS-PAGE analysis in panel (b). (b) SDS-PAGE analysis. MW, molecular weight marker; 1, culture filtrate; 2, 50% saturated ammonium sulfate precipitate; 3, 70% ethanol insoluble precipitate from 2; 4, 90% ethanol precipitate from supernatant of 70% ethanol suspension of 2; C, purified PaE*. DDW, deionized distilled water. *Preparation of purified PaE is described in S1 File.

## PaE precipitation by ethanol

After treating the culture filtrate of *P. antarctica* L1-S12 with various concentrations of ethanol, no precipitates were obtained in 0% or 25% (v/v) ethanol. Each precipitate obtained in 50%, 70%, and 90% (v/v) ethanol was dissolved in DDW, and the remaining enzyme was measured (Table 2). Only a small portion of PBSA-degrading activity precipitated in 50% or 70% (v/v) ethanol, whereas the entire activity was recovered in the precipitate generated in 90% (v/v) ethanol. The precipitate generated in 90% (v/v) ethanol solution prepared from the supernatant of the 70% (v/v) ethanol solution recovered nearly 90% of the activity in the original culture filtrate, and its specific activity rose to about 2.94-fold that of the original culture filtrate.

**Table 1. Survival of *Pseudozyma antarctica* cells after ethanol treatment.**

| Strain | Temperature (°C) | Ethanol (%) | Colony-forming ability on YM agar plate[a] | | | | | | Increment of cell turbidity in YM liquid culture[b] | | | | | |
|---|---|---|---|---|---|---|---|---|---|---|---|---|---|---|
| | | | Ethanol treatment (h) | | | | | | | | | | | |
| | | | 1 | 2 | 4 | 24 | 48 | 168 | 1 | 2 | 4 | 24 | 48 | 168 |
| GB-4(0) | 4 | 0 | + | + | + | + | + | + | + | + | + | + | + | + |
| | | 10 | + | + | + | + | + | + | + | + | + | + | + | + |
| | | 20 | + | + | ± | - | - | - | + | + | + | - | - | - |
| | | 25 | - | - | - | - | - | - | - | - | - | - | - | - |
| | | 30 | - | - | - | - | - | - | - | - | - | - | - | - |
| | | ≥40 | - | - | - | - | NT | NT | - | - | - | - | NT | NT |
| | 30 | 0 | + | + | + | + | + | + | + | + | + | + | + | + |
| | | 10 | + | + | + | - | - | - | + | + | + | ± | - | - |
| | | 20 | - | - | - | - | - | - | - | - | - | - | - | - |
| | | 25 | - | - | - | - | - | - | - | - | - | - | - | - |
| | | 30 | - | - | - | - | - | - | - | - | - | - | - | - |
| | | ≥40 | - | - | - | - | NT | NT | - | - | - | - | NT | NT |
| L1-S12 | 4 | 0 | + | + | + | + | + | + | + | + | + | + | + | + |
| | | 10 | + | + | + | + | + | + | + | + | + | + | + | + |
| | | 20 | + | + | + | - | - | - | + | + | + | + | - | - |
| | | 25 | - | - | - | - | - | - | - | - | - | - | - | - |
| | | 30 | - | - | - | - | - | - | - | - | - | - | - | - |
| | | ≥40 | - | - | - | - | NT | NT | - | - | - | - | NT | NT |
| | 30 | 0 | + | + | + | + | + | + | + | + | + | + | + | + |
| | | 10 | + | + | + | - | - | - | + | + | + | + | - | - |
| | | 20 | - | - | - | - | - | - | - | - | - | - | - | - |
| | | 25 | - | - | - | - | - | - | - | - | - | - | - | - |
| | | 30 | - | - | - | - | - | - | - | - | - | - | - | - |
| | | ≥40 | - | - | - | - | NT | NT | - | - | - | - | NT | NT |

[a] + (also indicated by grey shading of cells), *P. antarctica* colonies grew on the entire surface of the plate; ± (also indicated by light grey shading of cells), a few colonies grew on the plate; -, no colonies grew on the plate.

[b] + (also indicated by grey shading of cells), *P. antarctica* cells proliferated to $OD_{660} > 1.0$; ± (also indicated by light grey shading of cells), the cells proliferated to $0.05 < OD_{660} \leq 1.0$; -, the cells did not proliferate ($OD_{660} < 0.05$) in the liquid medium.

NT; not tested.

(a) (b)

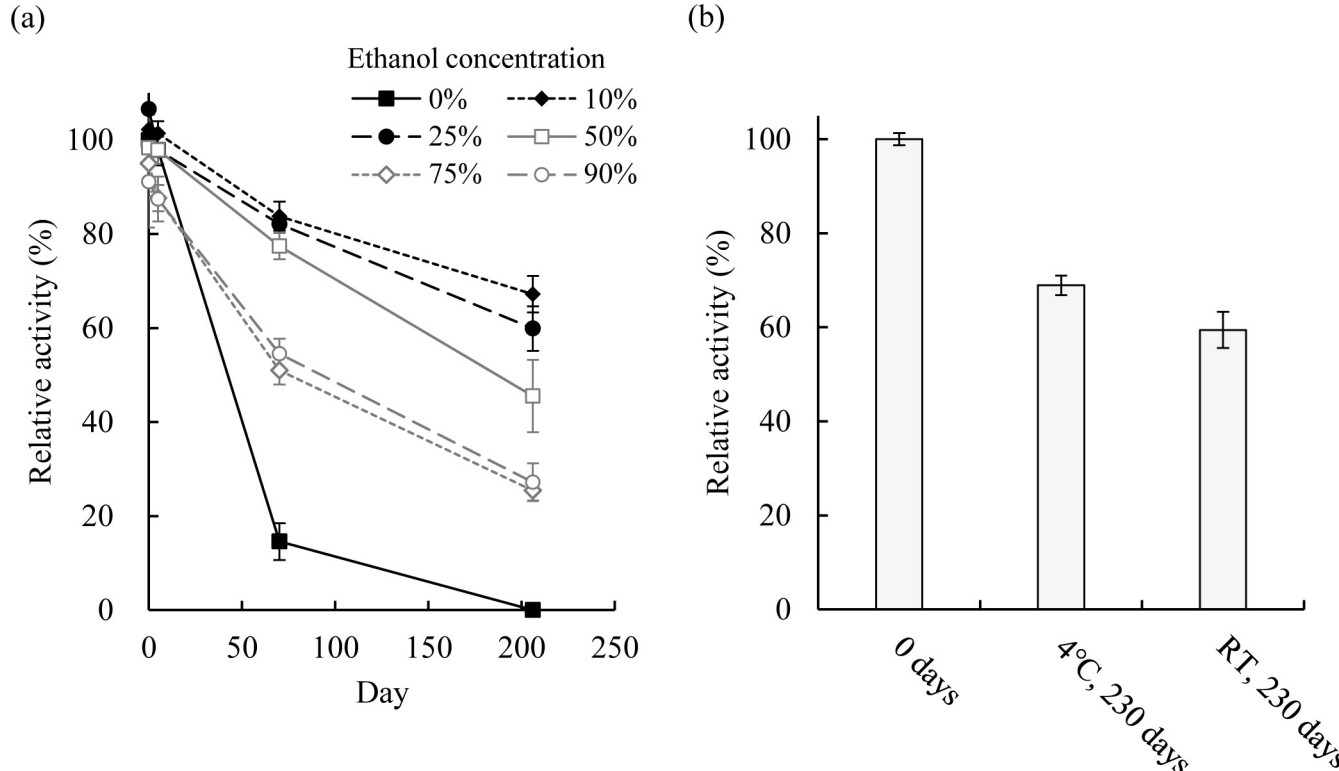

**Fig 2.** (a) Temporal changes in PaE activity in culture filtrates containing various amounts of ethanol. Ethanol was added to a final concentration of 0%, 10%, 25%, 50%, 75%, or 90% (v/v), and then the enzyme solution was stored at ambient temperature (RT) for up to 206 days. Relative activities are expressed as a percentage of the initial (0 day) activity of the control culture filtrate (containing 0% ethanol). Data are means ± standard deviation (SD) ($n = 3$). (b) Residual PaE activity in the solution of partially purified PaE containing 20% ethanol. Partially purified PaE obtained by preparation with the combination of ammonium sulfate precipitation and ethanol precipitation was dissolved in 20% ethanol and stored at 4°C or RT for 230 days. Relative activities are expressed as a percentage of the initial (0 day) activity. Data are presented as the mean ± SD ($n = 3$).

## Concentration of PaE from culture filtrates by sequential ammonium sulfate precipitation and ethanol precipitation

The PaE activity in the original culture filtrate of *P. antarctica* L1-S12 was 95% recovered in the precipitate generated by 50% saturated ammonium sulfate treatment (Table 3). This precipitate was then suspended in 70% (v/v) ethanol and centrifuged to remove insoluble material, the supernatant was collected, and ethanol was added to a final concentration of 90% (v/v). The PaE activity in the precipitate of the 90% (v/v) ethanol treatment was 91.2% of that

**Table 2. Concentration of PaE by ethanol precipitation.**

| Purification step | % recovery of PaE activity[a] | Protein concentration (mg/mL) | Precipitated protein (%)[a] | Specific activity |
|---|---|---|---|---|
| Culture filtrate | 100 | 4.84 | - | 1 |
| 50%-ethanol precipitation | 3.6 | 0.47 | 9.7 | 0.38 |
| 70%-ethanol precipitation | 10.5 | 0.90 | 18.5 | 0.57 |
| 90%-ethanol precipitation | 105.7 | 4.21 | 87.0 | 1.22 |
| 70%-ethanol soluble / 90%-ethanol precipitation[b] | 87.2 | 1.43 | 29.6 | 2.94 |

[a] Relative values compared with the original culture filtrate.

[b] Precipitation in the 90% (v/v) ethanol solution prepared from the supernatant of the 70% (v/v) ethanol solution.

**Table 3. Concentration of PaE by combination of ammonium sulfate precipitation and ethanol precipitation.**

| Purification step | % recovery of PaE activity[a] | Protein concentration (mg/mL) | Precipitated protein (%)[a] | Specific activity |
|---|---|---|---|---|
| Culture filtrate | 100 | 4.84 | - | 1 |
| 50%-saturated ammonium sulfate precipitation | 95.0 | 4.15 | 75.3 | 1.26 |
| 70%-ethanol soluble/ 90%-ethanol precipitation | 91.2 | 1.47 | 26.7 | 3.42 |

[a] Relative value compared with the culture filtrate.

in the original culture filtrate, indicating that most of the activity in the ammonium sulfate pellet was recovered after ethanol treatment. Because the specific activity of the final precipitate rose to 3.42-fold that in the original culture filtrate (Table 3), and the intensity of the protein bands other than PaE on SDS-PAGE decreased (Fig 1B, lane 4) compared with both the original culture filtrate (Fig 1B, lane 1) and the 50% saturated ammonium sulfate precipitate (Fig 1B, lane 2), we will refer to the final precipitate as "partially purified PaE".

## Stability of partially purified PaE in ethanol

The PBSA-degrading activity of the partially purified PaE is shown in Fig 2B. It had preserved more than half of its initial activity for 230 days in 20% (v/v) ethanol solution: 68% ± 2% and 59% ± 4% of initial activity at 4°C and ambient temperature, respectively (Fig 2B).

## Discussion

When PaE is used for BP product degradation, it is desirable to be able to sterilize the culture media after enzyme production, concentrate the enzyme from a large amount of the culture supernatant, and both stabilize and store the enzyme for a long period of time. Here, we explored the possibility of using ethanol treatment to achieve these objectives and investigated the optimal conditions.

First, we examined the degree of inactivation of *P. antarctica* cells following treatment with various concentrations of ethanol. We found that the cells completely lost their proliferating ability within 1 h in 25% (v/v) ethanol at 4°C or 20% (v/v) ethanol at 30°C, respectively (Table 1). This result suggests that the ethanol tolerance of the basidiomycetous yeast *P. antarctica* is similar to that of common ascomycetous yeasts such as *Candida albicans*, *Kluyveromyces lactis*, *Saccharomyces cerevisiae*, and *Schizosaccharomyces pombe* [19–22]. Because our simple method of inactivating P. antarctica cells with relatively low concentrations of ethanol was capable of treating large volumes of solution while maintaining PaE activity, there was no need for more powerful methods such as high-temperature steam sterilization or filtration through a cell-impervious membrane.

Next, we explored ethanol-based methods for preserving PaE without proliferation of cultured microbes and contamination by foreign spoilage microbes. Without aseptic operations, the *P. antarctica* filtrate suffered severe microbial contamination and the PaE activity was rapidly lost (Fig 2A). On the other hand, addition of ethanol to a final concentration of ≥10% (v/v) thoroughly prevented microbial contamination during the 206-day examination period. After storage at ambient temperature for 206 days, the PaE activity of the samples with ethanol concentrations of 10% or 25% remained at 67% or 60% of the initial value, respectively, which was higher than the activity remaining at higher ethanol concentrations (Fig 2A). These results suggest that both inactivation of *P. antarctica* and preservation of PaE activity in the culture supernatant can be achieved by using low percentages of ethanol. Therefore, ethanol treatment would prevent leakage of cultured microbes into the environment when the enzyme solution

is used in practical situations to accelerate the degradation of BP products close to where the product is used.

The PaE activity in the culture filtrate was entirely recovered (105.7%) and concentrated (1.22-fold) by a single step of precipitation at an ethanol concentration of 90% (v/v) (Table 2). Therefore, it is obvious that PaE recovers its activity after treatment with high-concentration ethanol. However, this method increases the solution volume tenfold by adding a large amount of ethanol and is disadvantageous in practical use when concentrating PaE from large amounts of culture supernatant. PaE has been reported to be precipitated in 50% saturated ammonium sulfate solution without losing its PBSA-degrading activity [12, 13]. To increase the practicality of the method for large quantities of culture filtrate, we devised a novel concentration protocol in which ammonium sulfate precipitation was followed by two ethanol precipitation steps: (1) isolation of 70% (v/v) ethanol–soluble substances from the precipitate and (2) precipitation of these substances in 90% (v/v) ethanol (Fig 1A). In our experiment, the sequential ammonium sulfate precipitation and ethanol precipitation effectively recovered about 90% of the enzyme activity in the original solution (Table 3). The specific activity of the final product was 3.42-fold and 2.80-fold that of the original culture filtrate (Table 3) and the suspension after single-step precipitation with 90% (v/v) ethanol (Table 2), respectively. The reduction in intensity of non-specific protein bands on SDS-PAGE showed that the purity of the solution was increased compared with the original culture filtrate (Fig 1B). Because the 50% saturated ammonium sulfate precipitate was suspended in 70% (v/v) ethanol rather than DDW, the volume of ethanol added in the 90% (v/v) precipitation step was one-third of that required for single-step precipitation with ethanol. The combination of ammonium sulfate precipitation and ethanol precipitation can reduce both the ethanol usage and the size of the required equipment compared to the single-step precipitation with ethanol. In addition, partially purified PaE kept more than 60% of its PBSA-degrading activity in 20% (v/v) ethanol solution at ambient temperature for more than 7 months (Fig 2B). Thus, our method enables a large volume of PaE solution to be stored without special equipment. The ethanol treatment serves all the requirements for the preparation of enzyme.

## Conclusions

Ethanol treatment is an effective way to (a) eliminate contamination of the PaE solution by foreign spoilage microbes, (b) stabilize the enzyme activity in the process of concentration and storage of PaE, and (c) inactivate cultured microbes and prevent their escape into the external environment when PaE is used for BP product degradation. Concentration of PaE by sequential ammonium sulfate precipitation and ethanol precipitation reduces ethanol use and increases PaE purity when compared with concentration of the enzyme by ethanol precipitation alone.

## Supporting information

**S1 Raw image.**
(TIF)

**S2 Raw image.**
(TIF)

**S1 File. SDS-PAGE analysis of the purified PaE.** MW, molecular weight marker (Precision Plus Protein Unstained Standards, Bio-Rad Laboratories Inc.); 1, culture filtrate of *Pseudozyma antarctica* X-14 strain; 2, purified PaE that was eluted from a Mono S column (GE Healthcare Life Sciences) using a linear gradient of 0–0.2 M NaCl and concentrated with an Amicon Ultra

Centrifugal Filter (pore size: 3 kDa; Merck KGaA).
(DOCX)

## Acknowledgments

We thank Mrs. Xiao-hong Cao and Mrs. Keiko Masunaga of NIAES for their technical assistance.

## Author Contributions

**Conceptualization:** Takumi Tanaka, Ken Suzuki, Hiroko Kitamoto.

**Formal analysis:** Takumi Tanaka, Ken Suzuki, Hirokazu Ueda, Yuka Sameshima-Yamashita.

**Funding acquisition:** Hiroko Kitamoto.

**Investigation:** Takumi Tanaka, Ken Suzuki, Hirokazu Ueda.

**Methodology:** Takumi Tanaka, Ken Suzuki, Hirokazu Ueda.

**Project administration:** Hiroko Kitamoto.

**Validation:** Takumi Tanaka, Ken Suzuki, Hirokazu Ueda, Yuka Sameshima-Yamashita, Hiroko Kitamoto.

**Writing – original draft:** Takumi Tanaka, Ken Suzuki, Hirokazu Ueda, Yuka Sameshima-Yamashita, Hiroko Kitamoto.

**Writing – review & editing:** Takumi Tanaka, Ken Suzuki, Hirokazu Ueda, Yuka Sameshima-Yamashita, Hiroko Kitamoto.

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
