## [Decision Letter · Decision Letter 0]

18 Mar 2021

PONE-D-20-38928

Ethanol treatment for sterilization, concentration, and stabilization of biodegradable plastic–degrading enzyme from Pseudozyma antarctica culture supernatant

PLOS ONE

Dear Dr. Hiroko Kitamoto,

Thank you for submitting your manuscript to PLOS ONE. After careful consideration, we feel that it has merit but does not fully meet PLOS ONE’s publication criteria as it currently stands. Therefore, we invite you to submit a revised version of the manuscript that addresses the points raised during the review process.

Manuscript PONE-D-20-38928 reports a new method for sterilization, concentration and stabilization of a plastic–degrading enzyme. In general the manuscript is well organized and clear. However, as recommended by both reviewers, the discussion section should be improved, with a comparative analysis of the data obtained with that already reported in the literature.

In addition, English must be improved before acceptance.

We look forward to receiving your revised manuscript.

Kind regards,

Olga C. Nunes

Academic Editor

PLOS ONE

Journal Requirements:

Reviewers' comments:

Reviewer's Responses to Questions

**Comments to the Author**

1. Is the manuscript technically sound, and do the data support the conclusions?

Reviewer #1: Yes

Reviewer #2: Yes

2. Has the statistical analysis been performed appropriately and rigorously? 

Reviewer #1: Yes

Reviewer #2: Yes

3. Have the authors made all data underlying the findings in their manuscript fully available?

Reviewer #1: Yes

Reviewer #2: Yes

4. Is the manuscript presented in an intelligible fashion and written in standard English?

Reviewer #1: No

Reviewer #2: Yes

5. Review Comments to the Author

Reviewer #1: The overall study seems interesting, however, few queries needs to be addressed.

1. The grammar is poor. The manuscript needs extensively revised.

2. Reframe the abstract.

3. Correct line no. 50

4. Edit line no. 130

5. Edit line no. 269

6. Edit line no. 282

7. Cite some more references

Reviewer #2: It is preferable that the discussion section should include the comparative analysis of the specific activity and yield of the enzyme precipitated by the usual method and this method, in order to highlight the advantage / importance of this research work. The entire manuscript talks about this research study but it is good to compare the work with that of previous reports, so that it would give a more clear idea for the readers.

6. PLOS authors have the option to publish the peer review history of their article (what does this mean?). If published, this will include your full peer review and any attached files.

Reviewer #1: No

Reviewer #2: **Yes: **Subathra Devi C

---

## [Author Response · Author response to Decision Letter 0]

19 Apr 2021

[Response to Reviewer #1]

The overall study seems interesting, however, few queries needs to be addressed.

→ We thank the reviewers for their helpful comments and suggestions. Below are our point-by-point responses to the reviewer’s comments.

1. The grammar is poor. The manuscript needs extensively revised.

→ We have modified the entire manuscript. The manuscript has been edited carefully by two native English-speaking professional editors from ELSS, Inc. (elss@elss.co.jp, http://www.elss.co.jp). All changes are highlighted in yellow.

2. Reframe the abstract.

→ We have edited the abstract to reflect the content of the manuscript more accurately as follows:

Line nos. 24–42 (revised manuscript):

Biodegradable plastics must be sufficiently stable to maintain functionality during use but need to be able to degrade rapidly after use. We previously reported that treatment with an enzyme named PaE, secreted by the basidiomycete yeast Pseudozyma antarctica can speed up this degradation. To facilitate the production of large quantities of PaE, here, we aimed to elucidate the optimal conditions of ethanol treatment for sterilization of the culture supernatant and for concentration and stabilization of PaE.

The results showed that Pseudozyma antarctica completely lost its proliferating ability when incubated in ≥20% (v/v) ethanol. When the ethanol concentration was raised to 90% (v/v), PaE formed a precipitate; however, its activity was restored completely when the precipitate was dissolved in water. To reduce ethanol use, PaE was successfully concentrated and recovered by sequential ammonium sulfate precipitation and ethanol precipitation steps. Over 90% of the activity in the original culture supernatant was recovered and the specific activity was increased 3.4-fold. By preparing the enzyme solution at a final concentration of 20% (v/v) ethanol, about 60% of the initial activity was maintained at ambient temperature for over 6 months without growth of microbes. We conclude that ethanol treatment is effective for sterilization, concentration, and stabilization of PaE, and that concentrating PaE by sequential ammonium sulfate precipitation and ethanol.

3. Correct line no. 50

→ We have corrected this sentence, and have added references. 

Line nos. 45–47 (revised manuscript):

Since large-scale production of the plastics started in the 1950’s, more than 8 billion tons of plastic products have been used worldwide, and more than 6 billion tons have been disposed of as waste [1,2].

In addition, we have edited the Introduction (see changes highlighted in yellow in the manuscript file)

4. Edit line no. 130

→ We have modified this sentence to describe in more detail how the concentration of purified PaE was calculated: 

Line nos. 131–139 (revised manuscript):

The concentration of purified PaE was calculated as described previously [13]. In short, the purified PaE solution was appropriately diluted, and its absorbance was measured at 280 nm (A280). Then, the concentration of PaE was calculated according to the following equation: A280 (M−1 cm−1) = 5,500nW + 1,490nY + 125nC [17,18], where M (=20,362.41) is the average molecular weight calculated using GENETYX software ver. 9 (GENETYX, Tokyo, Japan), nW is the number of Trp residues (=1), nY is the number of Tyr residues (=7), and nC is the number of disulfide bonds (=2), per polypeptide chain based on the mature amino acid sequence of PaE [6].

5. Edit line no. 269

→ We have edited line no 269 and have added a new paragraph above it as follows:

Line nos. 278–284 (revised manuscript):

When PaE is used for BP product degradation, it is desirable to be able to sterilize the culture media after enzyme production, concentrate the enzyme from a large amount of the culture supernatant, and both stabilize and store the enzyme for a long period of time. Here, we explored the possibility of using ethanol treatment to achieve these objectives and investigated the optimal conditions. 

First, we examined the degree of inactivation of P. antarctica cells following treatment with various concentrations of ethanol. 

In addition, we have edited the Discussion (see changes highlighted in yellow in the manuscript file)

6. Edit line no. 282

→ We have edited line no. 282 and its surrounding lines as follows:

Line nos. 299–302 (revised manuscript):

After storage at ambient temperature for 206 days, the PaE activity of the samples with ethanol concentrations of 10% or 25% remained at 67% or 60% of the initial value, respectively, which was higher than the activity remaining at higher ethanol concentrations (Fig 2a).

7. Cite some more references

→ We have cited eight new references. In addition, we have corrected the reference style to match the style and format of PLOS ONE.

[Response to Reviewer #2]

It is preferable that the discussion section should include the comparative analysis of the specific activity and yield of the enzyme precipitated by the usual method and this method, in order to highlight the advantage / importance of this research work. The entire manuscript talks about this research study but it is good to compare the work with that of previous reports, so that it would give a more clear idea for the readers.

→ We thank the reviewers for their helpful comment and the suggestion. We tried to compare this work with that of previous reports. We previously reported purification of PaE from the culture filtrate of Pseudozyma antarctica (Kitamoto et al, 2011; reference no. 12 in the revised manuscript). In this report, enzyme-producing microbe can be removed from the culture supernatant by passage through a cell impermeable membrane using a dedicated facility, however, the concentration and preservation steps are still necessary. Therefore, we have added a new paragraph and new a sentence in the Discussion as follows: 

Line nos. 278–282 (revised manuscript):

When PaE is used for BP product degradation, it is desirable to be able to sterilize the culture media after enzyme production, concentrate the enzyme from a large amount of the culture supernatant, and both stabilize and store the enzyme for a long period of time. Here, we explored the possibility of using ethanol treatment to achieve these objectives and investigated the optimal conditions.

Line nos. 334–335 (revised manuscript):

The ethanol treatment serves all the requirements for the preparation of enzyme.

---

## [Decision Letter · Decision Letter 1]

24 May 2021

Ethanol treatment for sterilization, concentration, and stabilization of a biodegradable plastic–degrading enzyme from Pseudozyma antarctica culture supernatant

PONE-D-20-38928R1

Dear Dr. Kitamoto,

We’re pleased to inform you that your manuscript has been judged scientifically suitable for publication and will be formally accepted for publication once it meets all outstanding technical requirements.

Kind regards,

Olga C. Nunes

Academic Editor

PLOS ONE

Additional Editor Comments (optional):

Reviewers' comments:

Reviewer's Responses to Questions

**Comments to the Author**

1. If the authors have adequately addressed your comments raised in a previous round of review and you feel that this manuscript is now acceptable for publication, you may indicate that here to bypass the “Comments to the Author” section, enter your conflict of interest statement in the “Confidential to Editor” section, and submit your "Accept" recommendation.

Reviewer #1: All comments have been addressed

Reviewer #2: All comments have been addressed

2. Is the manuscript technically sound, and do the data support the conclusions?

Reviewer #1: Yes

Reviewer #2: Yes

3. Has the statistical analysis been performed appropriately and rigorously? 

Reviewer #1: Yes

Reviewer #2: Yes

4. Have the authors made all data underlying the findings in their manuscript fully available?

Reviewer #1: Yes

Reviewer #2: Yes

5. Is the manuscript presented in an intelligible fashion and written in standard English?

Reviewer #1: Yes

Reviewer #2: Yes

6. Review Comments to the Author

Reviewer #1: Authors have addressed all the comments. The manuscript can be accepted now in the current form. No further comment

Reviewer #2: (No Response)

7. PLOS authors have the option to publish the peer review history of their article (what does this mean?). If published, this will include your full peer review and any attached files.

Reviewer #1: No

Reviewer #2: **Yes: **Subathra Devi C

---

## [Editor Report · Acceptance letter]

26 May 2021

PONE-D-20-38928R1 

Ethanol treatment for sterilization, concentration, and stabilization of a biodegradable plastic–degrading enzyme from *Pseudozyma antarctica* culture supernatant 

Dear Dr. Kitamoto:

I'm pleased to inform you that your manuscript has been deemed suitable for publication in PLOS ONE. Congratulations! Your manuscript is now with our production department. 

Kind regards, 

on behalf of

Dr. Olga Cristina Pastor Nunes 

Academic Editor

PLOS ONE